# Comparative Analysis of Water Extracts from Roselle (*Hibiscus sabdariffa* L.) Plants and Callus Cells: Constituents, Effects on Human Skin Cells, and Transcriptome Profiles

**DOI:** 10.3390/ijms241310853

**Published:** 2023-06-29

**Authors:** Won Kyong Cho, Soo-Yun Kim, Sung Joo Jang, Sak Lee, Hye-In Kim, Euihyun Kim, Jeong Hun Lee, Sung Soo Choi, Sang Hyun Moh

**Affiliations:** 1College of Biotechnology and Bioengineering, Sungkyunkwan University, Suwon 16419, Republic of Korea; wonkyong@gmail.com; 2Plant Cell Research Institute of BIO-FD&C Co., Ltd., Incheon 21990, Republic of Korea; sykim@biofdnc.com (S.-Y.K.); sjjang@biofdnc.com (S.J.J.); slee@biofdnc.com (S.L.); hikim@biofdnc.com (H.-I.K.); ehkim@biofdnc.com (E.K.); jhlee@biofdnc.com (J.H.L.); 3Daesang Holdings, Jung-gu, Seoul 04513, Republic of Korea; schoi@daesang.com

**Keywords:** Roselle, water extracts, cosmetics, skin cells, transcriptome profiling

## Abstract

Roselle (*Hibiscus sabdariffa* L.) is a plant that has traditionally been used in various food and beverage products. Here, we investigated the potential of water extracts derived from Roselle leaves and callus cells for cosmetic and pharmaceutical purposes. We generated calluses from Roselle leaves and produced two different water extracts through heat extraction, which we named Hibiscus sabdariffa plant extract (HSPE) and Hibiscus sabdariffa callus extract (HSCE). HPLC analysis showed that the two extracts have different components, with nucleic acids and metabolites such as phenylalanine and tryptophan being the most common components in both extracts. In vitro assays demonstrated that HSCE has strong anti-melanogenic effects and functions for skin barrier and antioxidant activity. Transcriptome profiling of human skin cells treated with HSPE and HSCE showed significant differences, with HSPE having more effects on human skin cells. Up-regulated genes by HSPE function in angiogenesis, the oxidation-reduction process, and glycolysis, while up-regulated genes by HSCE encode ribosome proteins and IFI6, functioning in the healing of radiation-injured skin cells. Therefore, we suggest that the two extracts from Roselle should be applied differently for cosmetics and pharmaceutical purposes. Our findings demonstrate the potential of Roselle extracts as a natural source for skincare products.

## 1. Introduction

Roselle (*Hibiscus sabdariffa* L.), also known as red sorrel, is a member of the genus *Hibiscus* in the family Malvaceae [1]. It is native to Africa and is widely cultivated in tropical and subtropical regions, including India, Indonesia, and Malaysia [2]. The plant is best known for its distinctive red flower, which is composed of cup-shaped calyces. The thick and fleshy calyces of the Roselle flower have been used worldwide as a raw material for a variety of food and beverage products, including popular cold and hot drinks, jams, syrups, and wines, for a considerable period of time [3]. Additionally, the calyces of the Roselle plant are frequently employed as natural food colorants [4].

Several previous studies have identified metabolites present in various tissues of the Roselle plant [5]. For instance, the major organic acids found in the aqueous extract of Roselle flowers are citric, malic, and tartaric acids [6]. Interestingly, the concentration of these organic acids increases during the development of the calyces but decreases as the calyces ripen. In a previous study, TLC and HPLC fingerprint analyses of methanol extracts from Roselle flowers revealed several flavonoids, including quercetin, luteolin, gossypetin, and hibiscetin, as well as other known chemical components such as anthocyanins, including delphinidin-3-glucosyl-xyloside (hibiscin) and cyanidin-3-glucosylxyloside, phenols and phenolic acids like protocatechuic and pelargonidic acid, eugenol, plant sterols, and ergosterol [7].

For a long time, Roselle flower extracts have been used in traditional medicine to treat fever, high blood pressure, and liver diseases [5]. Numerous recent studies have revealed the many pharmacological characteristics of Roselle flower extract. For instance, the leaves and flowers of Roselle are rich in several anthocyanins that provide antioxidant activities and have been used as a diuretic and sedative [8]. Moreover, anthocyanins derived from Roselle demonstrated anticancer activity against leukemia in both rats and humans [9]. Additionally, extracts of Roselle flowers have shown antipyretic, antinociceptive, anti-inflammatory activities, and anticholesterol effects [9].

Although many plants produce phytochemical constituents, such as secondary metabolites, that are useful in pharmaceutics and cosmetics, obtaining high amounts of these compounds directly from plants is often difficult due to the low amounts produced. To obtain natural compounds in high quantities, plant cell culture technology is frequently used to produce plant-derived metabolites [10]. This technology allows for the production of secondary metabolites in vitro without the obstacles posed by environmental factors and geographical regions.

The transcriptome refers to the collection of all RNA molecules transcribed from the genome [11]. Thanks to the rapid development of high-throughput sequencing and bioinformatic tools, it is now possible to examine the effects of plant extracts on specific human cell lines [12]. Several previous studies have demonstrated the usefulness of RNA sequencing in identifying human RNAs regulated by a specific plant extract [13,14,15].

In this study, we selected the leaves of Roselle because generating calluses from Roselle flowers can be more challenging compared to using leaves. Callus formation depends on various factors, including the tissue’s ability to undergo dedifferentiation and reprogram into a totipotent state [16]. In many plant species, including Roselle, leaves generally have a higher dedifferentiation capacity compared to other plant organs. Flower tissues, such as petals or reproductive structures, have a specialized cellular composition and are more differentiated than leaves. These specialized cells may be less responsive to in vitro culture conditions and callus induction due to developmental changes. Therefore, we decided to study leaves as they have a higher propensity for callus formation in this study.

In this study, we generated plant cells from the leaves of Roselles and propagated them using a bioreactor. We prepared extracts from the plant cells and Roselles leaves using a hot water extraction method and compared the constituents and transcriptomes between the two water extracts.

## 2. Results

### 2.1. Generation and Propagation of Plant Cells from Roselle Leaves

Roselle seeds were sterilized using EtOH and NaOCl (Figure 1A) and germinated on the growth medium (Figure 1B). The small pieces of leaves from the Roselle seedlings were used for the induction of Roselle callus (Figure 1C,D). The Roselle callus was induced and propagated on the Murashige and Skoog (MS) medium (Figure 1E,F). Mature Roselle calluses was used for the suspension cell culture using a bioreactor.

Next, we prepared two different water extracts from Roselle leaves and plant cells through heat extraction. The heat extraction method was chosen based on the intended application of Roselle extract, which is being developed as a sterilized additive for inclusion in cosmetics and non-prescription pharmaceuticals. The selection of this method aligns with the manufacturing process of these products and consequently led to the utilization of high-pressure sterilization extraction. For simplicity, we named two extracts: *Hibiscus sabdariffa* plant extract (HSPE) and *Hibiscus sabdariffa* callus extract (HSCE).

### 2.2. Identification of Components in Water Extracts of Hibiscus sabdariffa Using High-Performance Liquid Chromatography

We analyzed the components present in two water extracts, HSPE and HSCE, using high-performance liquid chromatography (HPLC). Chromatograms of the two extracts were obtained at three different wavelengths (210, 255, and 305 nm) (Figure 2), which revealed the differences between HSPE (Figure 2A,C) and HSCE (Figure 2D,F). Although not all peaks from HSPE were identified, we were able to detect guanine, phenylalanine, and tryptophan from both HSPE and HSCE at 210 and 255 wavelengths (Figure 2A,B). Conversely, adenine, uridine, adenosine, and guanosine, which are components of nucleic acids, were only identified from HSCE (Figure 2D,E). Interestingly, the identified peaks from HSCE appeared faster than those from HSPE in the chromatograms, indicating that the components of HSCE are more hydrophilic than those of HSPE.

### 2.3. Assessment of the Cosmetical Effects of HSCE through In Vitro Studies

We evaluated the effect of HSCE on cell cytotoxicity using the CCK-8 assay. After treatment with three different concentrations of HSCE on HaCaT cells, the cell viability was measured compared to the control treated with sterile water (Figure 3A). As the concentration of HSCE increased from 1% to 10%, the cell viability decreased from 85.56% to 76.16%.

The melanin inhibitory activity was measured after treatment with different concentrations of HSCE on B16F1 cells (mouse melanoma cell line) (Figure 3B). Compared to the positive control treated with 100 ppm kojic acid (36.24% of melanin inhibitory activity), all HSCE-treated samples showed a significantly lower melanin inhibitory activity ranging from 5.81% to 11.48%, showing statistical difference with the positive control.

Filaggrin (FLG) is a structural protein required for the development and maintenance of the skin barrier. Using the expression of FLG, we measured the skin barrier effect of HSCE after treatment with three different HSCE concentrations on HaCaT cells (Figure 3C). The relative expression of the FLG gene in the positive control sample treated with 1% glyceryl glucoside was 3.71, while the expression of the FLG gene for 1%, 5%, and 10% of HSCE-treated samples was 0.63, 2.06, and 5.57. Compared to the control treated with sterile water, the positive control and 10% of HSCE-treated samples showed significant differences supported by statistical results (*p* < 0.01 and *p* < 0.001).

The antioxidant effect of HSCE was measured based on the expression of SOD1 after treatment with different HSCE concentrations on HaCaT cells (Figure 3D). Compared to the control treated with sterile water, the positive control showed significant up-regulation of SOD1 gene expression (1.13). Among the three different HSCE concentrations, only 10% of HSCE (1.47) showed a significant difference compared to the control, although the expression of SOD1 was increased after treatment with 5% and 10% of HSCE.

### 2.4. Transcriptomic Analysis of HaCaT Cells in Response to Extracts of HS Derived from Plant and Callus

To investigate the transcriptional changes in HaCaT cells in response to HS extracts, we performed RNA sequencing. We collected a total of nine HaCaT samples from three different experimental conditions: a control group treated with distilled water, a group treated with HSPE (HS extract from the plant), and a group treated with HSCE (HS extract from the callus). Each condition consisted of three biological replicates (Table 1).

We treated HaCaT cell samples with either distilled water (control) or extracts of HS derived from the plant and callus. Each condition included three different biological replicates. HS refers to *Hibiscus sabdariffa*.

Total RNA was extracted from each sample and used to prepare a library for RNA sequencing. The library was sequenced in paired-end mode using the NovaSeq 6000 system. The total number of sequenced reads ranged from 10 gigabase pairs (HSPE-R1) to 13 gigabase pairs (HSCE-R3), while the total number of reads in each library ranged from 99 million (HSPE-R1) to 131 million (HSCE-R3) (Table 2). The Q20 and Q30 quality scores indicated high sequencing read quality.

### 2.5. Identification of Differentially Expressed RNAs in Response to HS Extract Treatment

We filtered out poor-quality reads and mapped the remaining clean reads from each library onto the human genome using BWA with default parameters. The eXpress program was used to calculate the number of mapped reads for each RNA (transcript) by analyzing the aligned SAM files. We then used the DESeq2 program to analyze the differentially expressed RNAs (DERs) by comparing the extract-treated condition to the control condition. We identified DERs for HSPE, HSCE, and HS-All (comparison of all treated samples to control samples) using an adjusted *p*-value less than 0.01 and a log10-converted fold change greater than 2 as a cutoff. The volcano plots (Figure 4A–C) indicated that HSPE had a large number of DERs compared to HSCE. However, when we compared all extract-treated samples to the control samples, we did not identify any DERs (Figure 4C). We identified several transcript variants encoded by the same gene, and for simplicity, we regarded each transcript variant as a single DER. We identified 111 DERs from HSPE (Appendix A), which were further divided into 92 up-regulated RNAs and 19 down-regulated RNAs (Figure 4D). In the case of HSCE, we identified five up-regulated and one down-regulated RNA (Figure 4D).

Of the identified DERs, the majority were mRNAs, but we also found 10 noncoding RNAs (ncRNAs), including WDR77, MT1L, CLCA4, FGF11, SDAD1P1, MIR210HG, LOC107987250, LOC101930275, LOC101930275, and EGLN3-AS1. All of the ncRNAs, except for WDR77, were up-regulated, with EGLN3-AS1, which encodes EGLN3 anti-sense RNA 1, being the most highly up-regulated transcript. Interestingly, we also observed that 39 DERs were unique transcripts encoded by a single gene, while 72 DERs were RNA variants. For example, we identified six DERs encoded by the NDRG1 gene, which encodes N-myc downstream regulated 1, and six DERs encoded by the SAMD4A gene, which encodes the sterile alpha motif domain containing 4A. In total, we identified 12 down-regulated genes and 51 up-regulated genes following HSPE treatment. The representative highly up-regulated RNAs were BNIP3, RNASE7, MIR210HG, SLC6A8, PPP1R3C, and SAMD4A, while the highly down-regulated RNAs were AKR1B10, NME1, and ALDH1A (Table 3).

### 2.6. Functional Classification of Differentially Expressed RNAs Associated with HSPE Treatment

Next, we examined the enriched functions of up- and down-regulated DERs associated with HSPE across seven different databases (Appendix A and Figure 5A). Based on the gene ontology (GO) term enrichment analysis, we identified 45 GO terms for up-regulated DERs and 3 GO terms for down-regulated DERs according to biological processes. According to the cellular components, the number of enriched GO terms for up-regulated DERs was lower than that for down-regulated DERs. Regarding the molecular function, 48 and 41 enriched GO terms were identified for up- and down-regulated DERs, respectively. The enriched functions for down-regulated DERs were higher than those for up-regulated DERs across three databases such as Panther, Reactome, and WikiPathways, except for KEGG. When comparing the list of enriched functions for the two groups, only two functions, mitochondrion (GO:0005739) and fructose and mannose metabolism (hsa00051), were commonly identified for both up- and down-regulated DERs (Figure 5B).

A total of 51 up-regulated genes were assigned to known GO terms, as shown in Figure 6A. Among the assigned functions, the metabolic process (34 genes), biological regulation (27 genes), and response to stimulus (25 genes) were frequently identified according to the biological process. Many up-regulated genes were targeted to the membrane (19 genes), nucleus (16 genes), and membrane-enclosed lumen (16 genes). Additionally, some up-regulated genes were associated with the microbody (one gene) and endosome (one gene). The most abundant molecular functions for up-regulated genes were protein binding (33 genes), ion binding (17 genes), and hydrolase activity (10 genes). Furthermore, molecular functions such as the translation regulator activity, lipid binding, chromatin binding, molecular transducer activity, and electron transfer activity were exclusively identified in up-regulated genes.

A total of 12 genes were assigned to known GO terms as down-regulated genes by HSPE treatment (Figure 6B). According to the biological process, the metabolic process (10 genes) and biological regulation (nine genes) were frequently identified, whereas only five genes were assigned to the response to stimulus. Many down-regulated genes were targeted to three cellular components: cytosol (seven genes), nucleus (seven genes), and membrane-enclosed lumen (seven genes). As compared to up-regulated genes, down-regulated genes were highly identified in the cytoskeleton (five genes) and extracellular space (four genes). According to the molecular functions, down-regulated genes were mostly associated with protein binding (nine genes), nucleotide binding (seven genes), and ion binding (six genes).

Next, we performed hierarchical clustering analysis of 111 genes in three different conditions. The heatmap revealed that 111 genes could be categorized into two distinct groups: Group I and Group II (Figure 7A). Group I consisted of 19 genes that showed down-regulation in both HSPE and HS-All conditions, but their expression levels remained unchanged in the HSCE condition. In contrast, Group II comprised 92 genes that showed up-regulation in both HSPE and HS-All conditions, but their expression levels were not significantly altered in the HSCE condition. In the HSCE condition, we identified five up-regulated and one down-regulated gene (Figure 7B).

### 2.7. Functional Categorization of Genes Up-Regulated by HSPE

We performed GO term enrichment analysis to identify the enriched functions of 51 up-regulated genes regulated by HSPE (Appendix A). Among the 45 enriched GO terms associated with the biological process, seven GO terms including the response to hypoxia (GO:0001666), carboxylic acid metabolic process (GO:0019752), oxidation-reduction process (GO:0055114), angiogenesis (GO:0001525), and female pregnancy (GO:0007565) were found to be significantly enriched (Appendix A and Figure 8A). Eight genes—*EGLN1*, *LOXL2*, *EGLN3*, *ADM*, *STC2*, *NDRG1*, *HK2*, *PDK1*, and *BNIP3*—were involved in response to hypoxia, while six genes—*HDAC5*, *EGLN1*, *PTPRB*, *LOXL2*, *ITGA5*, *ADM*, and *HK2*—were associated with angiogenesis (Appendix A). In terms of cellular components, 31 GO terms were enriched, with many up-regulated genes targeted to synapse (GO:0045202), sarcoplasm (GO:0016528), mitochondrion (GO:0005739), perinuclear region of cytoplasm (GO:0048471), and tertiary granule (GO:0070821) (Appendix A and Figure 8B). For instance, seven genes—*STXBP1*, *EGLN1*, *HAP1*, *ITGA5*, *SAMD4A*, *NDRG1*, and *BNIP3*—were targeted to synapse, while *FABP3* and *HK2* were associated with sarcoplasm (Appendix A). Of the identified 48 enriched GO terms according to molecular function, GO terms associated with monosaccharide binding (GO:0048029), carbohydrate binding (GO:0030246), and carboxylic acid binding (GO:0031406), and translation repression activity (GO:0030371) were significantly enriched (Figure 8C). Five genes—*EGLN1*, *EGLN3*, *P4HA1*, *HK2*, and *SLC2A3*—were involved in monosaccharide binding, while *NANOS1* and *SAMD4A* were involved in translation repressor activity.

Furthermore, we identified 10 enriched KEGG pathways, among which the pathway associated with glycolysis and gluconeogenesis (hsa00010) was the most significantly enriched (Appendix A). Using the Panther database, we found three pathways, including fructose and galactose metabolism (P02744), glycolysis (P00024), and the hypoxia response via HIF activation (P00030). Additionally, we identified nine enriched functions using the Reactome database, with glycolysis (R-HSA-70171), elastic fiber formation (R-HSA-1566948), and collagen formation (R-HSA-1474290) being significantly enriched. Finally, using the WikiPathway database, we found seven enriched functions, including glycolysis and gluconeogenesis (WP534) and photodynamic therapy-induced HIF-1 survival signaling (WP3614).

### 2.8. Functional Classification of Genes Down-Regulated by HSPE Treatment

We performed GO term enrichment analysis to identify enriched functions of 12 down-regulated genes in response to HSPE. Only three enriched GO terms associated with the biological process were identified, including the small-molecule metabolic process (GO:0044281), gland development (GO:0048732), and microtubule cytoskeleton organization (GO:0000226) (Appendix A and Figure 9A). On the other hand, 43 enriched GO terms were identified according to the cellular component. Among them, four genes, namely *NME1*, *CENPE*, *KIF20A*, and *AURKA*, were involved in polymeric cytoskeletal fiber (GO:0099513), while *SLC7A11* was associated with astrocyte projection (GO:0097449) (Appendix A and Figure 9B). Regarding molecular functions, 41 enriched GO terms were identified (Appendix A). Among them, genes associated with retinal dehydrogenase activity (GO:0001758), sulfur amino acid transmembrane transporter activity (GO:0000099), arginine N-methyltransferase activity (GO:0016273), and ATP binding (GO:0005524) were down-regulated (Figure 9C). Of these, *AKR1B10* and *ALDH1A1* were involved in retinal dehydrogenase activity, while five genes, namely *NME1*, *CENPE*, *KIF20A*, *EARS2*, and *AURKA*, were involved in ATP binding (Appendix A).

In addition, we identified seven metabolic pathways enriched by HSPE treatment using the KEGG database, including folate biosynthesis (hsa00790) and galactose metabolism (hsa00052) (Appendix A). According to the Panther pathway, five pathways were enriched, including de novo pyrimidine ribonucleotides biosynthesis (P02740) and heme biosynthesis (P02746). Using the Reactome database, a total of 25 pathways were identified, including kinesins (R-HSA-983189), COPI-dependent Golgi-to-ER retrograde traffic (R-HSA-6811434), and fructose catabolism (R-HSA-70350). Among the 14 pathways identified by the WikiPathway, we found folate-alcohol and cancer pathway hypotheses (WP1589), endoderm differentiation (WP2853), and phytochemical activity on NRF2 transcriptional activation (WP3).

### 2.9. Identification of Differentially Expressed RNAs Associated with HSCE

We identified six differentially expressed RNAs (DERs) after HSCE treatment. Interestingly, all five up-regulated DERs were 18S ribosomal RNAs (N1 to N5) (Table 4). Only a single gene, *IFI6* encoding interferon alpha inducible protein 6, was identified as a DER.

### 2.10. Confirmation of RNA Sequencing Results by Real Time RT-PCR

To validate the results obtained from RNA sequencing, we performed real-time RT-PCR analysis on 10 selected genes that exhibited differential expression (refer to Table 5). Among these genes, five were up-regulated by HSPE treatment, while one gene was up-regulated by HSCE treatment. Additionally, four genes, namely *ALDH1A1*, *NME1*, *AKR1B10*, and *IFI6*, were found to be down-regulated.

Comparative analysis between real-time RT-PCR and RNA sequencing results revealed that the expression patterns of all nine genes were consistent, except for the *ALDH1A1* gene (Figure 10). Interestingly, the *ALDH1A1* gene showed down-regulation in the RNA sequencing data, whereas it was up-regulated in the real-time RT-PCR results. Furthermore, out of the 10 genes analyzed, real-time RT-PCR results indicated significant differences between the control and treatment groups for eight genes, as supported by statistical tests.

## 3. Discussion

In this study, we generated a callus from Roselle leaves and further subjected it to mass production using a bioreactor. The cultured Roselle plant cells were extracted using water and heat. Using HPLC, diverse in vitro assays, and transcriptome analyses, we characterized the effects of Roselle plant cell extracts.

Previous studies have generated calluses from Roselle seedlings and demonstrated the enrichment of anthocyanins in Roselle calluses [17,18]. Hormone combinations play particularly important roles in callus growth and anthocyanin synthesis, which are inversely correlated [17]. Moreover, a previous study demonstrated that yeast extract induced anthocyanin production in the callus generated from Roselle seedlings [18]. It is also possible to select the highly anthocyanin-producing callus line with a red and purple aggregate from the Roselle cell line [17]. In this study, we established a mass cell culture of a generated Roselle callus using a bioreactor for the first time, which can be useful for commercial applications.

The most commonly identified metabolites from HSPE and HSCE were guanine, phenylalanine, and tryptophan. Guanine is one of the four main nucleobases that constitute nucleic acids such as DNA and RNA. Phenylalanine and tryptophan are aromatic amino acids required for protein synthesis. For example, phenylalanine is involved in the synthesis of several secondary metabolites such as plant pigments including flavonoids, isoflavonoids, and proanthocyanidins, while tryptophan is related to the synthesis of vitamins and plant hormones [19].

In this study, we used water and heat to obtain extracts from the Roselle plant and callus since the main purpose of this study is to apply the HSCE for cosmetics and pharmaceutical purposes. Normally, only 1% of extracts derived from diverse plant cells are commercially utilized. However, in this study, we investigated the effects of three different concentrations, specifically 1%, 5%, and 10%. The results from the cell viability assay conducted on HaCaT cells indicate that the water extract of roselle cells exhibited no cytotoxicity, irrespective of the HSCE concentration used. Furthermore, the water extract showed lower cytotoxicity compared to the control group treated with sterile water. Interestingly, the cell viability remained unaffected by the concentration of HSCE in the water extract of roselle cells. Moreover, HSCE showed strong anti-melanogenic effects by repressing the expression of α-melanocyte stimulating hormone (α-MSH). As the concentration of HSCE increased, the melanin inhibition activity increased.

FLG protein functions in the skin’s barrier. Thus, the high expression of the *FLG* gene suggests the high ability of the skin barrier. The expression of *FLG* was increased by the treatment of 5% and 10% of HSCE on HaCaT cells. The antioxidant activity of HSCE on HaCaT cells was measured by the expression of *SOD1*, which is an antioxidant enzyme catalyzing superoxide breakdown. It was observed that only the highest concentration of HSCE (10%) resulted in a significant difference (1.47) when compared to the control group. This implies that the 10% HSCE treatment exhibited a more pronounced antioxidant effect in terms of *SOD1* gene expression. However, it is worth noting that there was an overall increase in *SOD1* expression after treatment with both 5% and 10% HSCE, albeit without reaching statistical significance. These results suggest that HSCE, particularly at a concentration of 10%, has the potential to induce antioxidant effects through the modulation of *SOD1* gene expression in HaCaT cells.

To date, no studies have investigated the expression profiles of HaCaT cells in response to Roselle extracts. Our transcriptome analyses revealed that HSPE has a greater impact on HaCaT cells compared to HSCE. This result suggests that the difference in components between HSPE and HSCE, as demonstrated by HPLC analysis, may be responsible for the differential effects observed. However, we cannot determine exactly which component in the Roselle extract plays an important role in the change in HaCaT transcriptome. Hierarchical clustering analysis with 111 differentially expressed genes by HSPE showed that the expression of most differentially expressed genes by HSPE was not changed by HSCE treatment. This result suggests that some components exclusively present in HSPE might have a strong impact on the transcriptional changes in HaCaT cells.

The most significantly enriched functions for up-regulated genes by HSPE were associated with the response to hypoxia, the carbohydrate derivative biosynthetic process, the oxidation-reduction process, and angiogenesis. In fact, these four biological processes are highly connected. For example, hypoxia is defined as a condition in which there is a lack of adequate oxygen supply at a specific region of the body at the tissue level. It is known that many hypoxia-induced genes are involved in angiogenesis, which is a physiological process that forms new blood vessels from preexisting vasculature and occurs during development and tissue modeling [20]. For example, two hypoxia-inducible factors, *EGLN1* and *EGLN3*, which were up-regulated genes by HSPE in this study, participate in cell survival and adaptation upon diverse environmental oxygen levels by functioning in many biological processes such as angiogenesis, erythropoiesis, and tumorigenesis [21]. Other up-regulated genes involved in angiogenesis, such as *HDAC5*, *PTPRB*, *LOXL2*, *ITGA5*, *ADM*, and *HK2*, might also be involved in wound healing, growth, and female pregnancy [22,23]. For example, PTPRB, which encodes the receptor-type tyrosine phosphate beta (VE-PTP), is an enzyme specifically expressed in endothelial cells. VE-PTP plays important roles in vasculogenesis and blood vessel development [24] as well as for regulating adherens junction complex [25]. Furthermore, *LOXL2*, encoding lysyl oxidase homolog 2, is a secreted enzyme participating in extracellular matrix (ECM) crosslinking and in regulating angiogenesis through collagen IV scaffolding [26].

Three genes, namely *EGLN1*, *LOXL2*, and *HK2*, function in both angiogenesis and oxidation-reduction processes. It is known that those redox-signaling-associated genes play important roles in vascular angiogenesis [27]. Four up-regulated genes, namely *ITGA5*, *PSG5*, *ADM*, and *STC2*, in HSPE treatment, are involved in female pregnancy. Of these genes, STC2, encoding Stanniocalcin-2, is a secreted glycoprotein hormone family that is induced by diverse stresses, including hypoxia and nutrient deprivation [28]. Moreover, other enrichment analyses using different databases indicate that many up-regulated genes by HSPE are involved in hypoxia-inducible factor 1 (HIF-1) signaling pathway and glycolysis, such as fructose and mannose metabolism, as shown by GO term enrichment analysis. The activation of HIF-1-related signaling genes induced by HSPE suggests that HSPE can participate in cell survival, proliferation, angiogenesis, invasion, metastasis, cancer stemness, and metabolic reprogramming in human skin cells [29]. Furthermore, several studies suggest that natural products can affect the glycolysis pathway and can be used to inhibit cancer cells [30]. However, the possible role of HSPE with the suggested functions should be further characterized.

Many genes, including *COX6B2*, *ANKZF1*, *STXBP1*, *HAP1*, *DEPP1*, *P4HA1*, *HK2*, *PDK1*, and *BNIP3*, were up-regulated by HSPE, and they are targeted to the mitochondrion. *COX6B2* encodes cytochrome c oxidase subunit 6B2, which is involved in the metabolic reprogramming of several tumor cells by enhancing oxidative phosphorylation. Moreover, the expression of *COX6B2* is induced by hypoxia [31,32]. Seven up-regulated genes, such as *STXBP1*, *EGLN1*, *HAP1*, *ITGA5*, *SAMD4A*, *NDRG1*, and *BNIP3*, were targeted to synapses, which are structures that enable a nerve cell to pass an electrical or chemical signal to another neuron cell [33].

In comparison to HSPE, only a few genes, such as five 18S ribosomal genes and IFI6, were up-regulated by HSCE. The up-regulation of genes encoding components of the 18S ribosome indicates that HSCE promotes the synthesis of diverse proteins. *IFI6* encodes interferon alpha-inducible protein 6, which is induced by ionizing radiation and provides protection to skin cells that were injured by ionizing radiation [34,35]. Therefore, the treatment of HSCE in HaCaT cells might enhance protein synthesis and the healing of radiation-induced skin injury.

Significant differences were observed in the findings obtained from real-time RT-PCR analysis and transcriptome analysis using RNA sequencing when examining the effects of Roselle plant cell extracts. Specifically, real-time RT-PCR analysis yielded specific results related to cytotoxicity, FLG expression, SOD1 expression, and α-MSH expression following HSCE treatment. In contrast, transcriptome analysis provided comprehensive insights into the impact of HSPE and HSCE on HaCaT cells, highlighting variations in gene expression patterns and their connections to diverse biological processes. Real-time RT-PCR and RNA sequencing are complementary techniques, each with their own strengths and limitations. Due to disparities in sensitivity, a targeted versus unbiased approach, technical variability, data analysis, and biological variability, the specific outcomes derived from real-time RT-PCR may not directly align with those obtained through transcriptome analysis via RNA sequencing.

Taken together, HPLC analyses indicate that the components of HSPE and HSCE are different, although some components are commonly present in both extracts. Moreover, the differences in components between the two extracts resulted in different transcriptome profiles of HSPE and HSCE. Functional analyses of differentially expressed genes by the two extracts indicate that many up-regulated genes by HSPE were associated with hypoxia, angiogenesis, and glycolysis, while HSCE treatment induced five genes that were components of the 18S ribosome and IFI6 gene. Diverse in vitro assays indicate possible applications of HSCE for cosmetics associated with skincare. However, the potential functional roles of the identified genes associated with Roselle extracts should be further examined in the near future.

## 4. Materials and Methods

### 4.1. Induction and Propagation of Plant Cells from Roselle Leaves

The Roselle seeds used in this study (registration number Je-Chilgok-2017-1—01 ho) were purchased from an online shop called 1000 Seeds in 2019. To sterilize the seeds, 25 seeds were treated with 70% EtOH for 1 min, followed by 2.5% NaOCl for 20 min. This method is carefully designed to minimize any potential interference with the viability and activity of the extracts. Additionally, Tween-20 can be added to enhance the sterilization effect. After sterilization, the seeds are thoroughly rinsed with sterilized water multiple times to ensure the complete removal of the sterilization solution. The sterilized seeds were placed on filter paper to remove excess moisture and then germinated on a medium in a growth chamber at 24 °C ± 2 °C and 1000 Lux. The resulting seedlings were cut into pieces of 5 to 8 mm and transferred to Murashige and Skoog (MS) medium (Duchefa, Haarlem, The Netherlands) (Cat. No. M0222) containing MS (4.4 g/L), sucrose (30 g/L) (Duchefa) (Cat. No. S0809), gelrite (2.3 g/L) (Duchefa) (Cat. No. G1101), and auxin and cytokinin at pH 5.8. The initial explants were subcultured every two weeks on the same medium. After eight weeks, callus tissues derived from various media were evaluated based on their color, size, and growth rate. To induce callus formation, MS medium supplemented with 4.5 µM 2,4-D (Dichlorophenoxyacetic acid) (Duchefa) (Cat. No. D0911) was used. Cell lines suitable for suspension culture were selected by repeating the subculture process under the same conditions. Finally, a bioreactor at the Anti-Aging Research Institute of BIO-FD&C Co., Ltd., Incheon, Republic of Korea, was used for mass production of the Roselle cells.

### 4.2. Preparation of Two Different Roselle Extracts through Heat Extraction Method

To prepare two different Roselle extracts, HSPE was obtained from dried plant leaves, while HSCE was obtained from plant cells. Roselle leaves were first dried at 60 °C for two days, and then HSPE was extracted from the dried leaves using distilled water through heat extraction at 121 °C for 15 min. For HSCPE preparation, plant cells (callus) were induced from Roselle leaves and suspension cultured for six days. The plant cells were then harvested using a non-woven fabric filter and subjected to heat extraction using distilled water at 121 °C for 15 min. In both extract preparations, we used 2 g/L of the sample for extraction. After heat extraction, we removed the solids by filtration through a mesh.

### 4.3. HPLC Analysis of Water Extracts from Hibiscus sabdariffa

To compare the chromatographic data of the water extracts from HSPE and HSCE, we performed HPLC using an Agilent 1260 Infinity II system. Separation was achieved using a Shim-pack GIS C18 column (5 μm, 4.6 × 250 mm) (Shimadzu, Kyoto, Japan) maintained at 30 °C. The mobile phase was a mixture of solvent A (water containing 0.1% trifluoroacetic acid) and solvent B (acetonitrile containing 0.1% trifluoroacetic acid), which were pumped under the following gradient conditions: 0→5 min, 0% B→0% B; 5→75 min, 0% B→75% B; 75→77 min, 75% B→95% B; 77→82 min, 95% B→95% B; 82→84 min, 95% B→0% B and 84→95 min 0% B→0% B. The flow rate was set at 1.0 mL/min, and the injection volume was 20 μL. Chromatographic data were collected using a diode array detector (DAD) within the wavelength range of 200–600 nm, and the results were extracted at three characteristic wavelengths (210, 255, and 305 nm).

### 4.4. Culture and Treatment of Human Keratinocyte Cells with Roselle Extracts

To cultivate and differentiate human cells, primary cultured keratinocytes were isolated from human epidermal tissue and then seeded onto cell culture plates. Meanwhile, human HaCaT keratinocytes, which were obtained from American Type Culture Collection, were cultured in a dermal cell basal medium supplemented with 10% heat-inactivated fetal bovine serum (FBS) from Gibco (Carlsbad, CA, USA) and a 100 U/mL penicillin/streptomycin mixture from Lonza (Walkersville, MD, USA) at 37 °C in a humidified atmosphere containing 5% CO_2_. We conducted the experiments using HaCaT cells that had undergone passages ranging from 10 to 15 or fewer. Once the cells reached 80–90% confluence, they were cultured in the medium. The HaCaT cells were treated with HSCE, an extract derived from Roselle calluses, at concentrations of 1%, 5%, and 10% for a duration of 24 h. In this case, the 10% concentration corresponds to a final concentration of 0.02% when prepared at a ratio of 2 g/L. Evaluating the effects at this concentration allows for a suitable assessment of the extract’s potential effects while considering the practical application and potential dilution factors. The experiments included a control group treated with distilled water.

### 4.5. Assessment of HSCE Cell Cytotoxicity Using the CCK-8 Assay

The study aimed to assess the impact of HSCE on human skin cell (HaCaT) growth, propagation, and survival by conducting a CCK-8 assay. HaCaT cells were seeded at a density of 5 × 10^4^ cells per well in a 96-well plate and incubated for 24 h. The cells were then treated with final concentrations of 1%, 5%, and 10% of HSCE for 24 h, with sterile water serving as the control. After HSCE treatment, 1X CCK-8 solution (Cat. No. CCK-3000, Donginbio, Seoul, Republic of Korea) was added to each well, and the cells were incubated for 3 h. A Thermo Scientific Multiskan GO Microplate Spectrophotometer (Fisher Scientific Ltd., Vantaa, Finland) was used to measure the wavelength absorbance at 450 nm. The percentage of cell viability was calculated using the following formula: (absorbance of treated cells/absorbance of control cells) × 100.

### 4.6. Quantificiation of Melanin Content following HSCE Treatment

B16F1 cells (mouse melanoma cell line) were seeded at a density of 1 × 10^5^ cells per well in 6-well plates and incubated under cell culture conditions for 24 h. Afterward, the cells were treated with HSCE and cultured for an additional 72 h. A positive control of 100 ppm Kojic acid was also included, and 1 μM α-MSH was added to each sample to stimulate melanin production. After incubation, the top layer of the medium was removed, and the cells were harvested using 1X Trypsin-EDTA. A 300 μL solution of 1 N NaOH was added to the cell pellet, and the melanin was dissolved by boiling at 100 °C for 30 min. The dissolved melanin solution was then partially transferred onto a 96-well plate to measure the absorbance at 405 nm. The absorbance value of each sample was quantified based on the amount of protein, and the melanin production inhibition rate was expressed as a percentage. The calculation for the melanin production inhibition rate was as follows: Melanin production inhibition rate (%) = [1 − (test group melanin amount/α-MSH treated control melanin amount)] × 100.

### 4.7. Expression Analysis of FLG and SOD1 Marker Genes by Real-Time Reverse Transcription (RT-PCR)

To evaluate the impact of HSCE treatment on the expression levels of FLG and SOD1 genes, HaCaT cells were seeded at a density of 5 × 10^4^ cells per well and treated separately with three different concentrations of HSCE (1%, 5%, and 10%), with sterile water serving as the control. After treatment, RNA isolation and cDNA synthesis were performed using the SuperPrep™ Cell Lysis & RT Kit for qPCR (Cat. No. SC-101, Toyobo, Osaka, Japan) according to the manufacturer’s instructions. Real-time reverse transcription PCR (RT-PCR) was conducted using known primers for filaggrin (FLG) (Cat. No. QT02448138, Qiagen, Hilden, Germany) as a skin barrier marker and for superoxide dismutase 1 (SOD1) (Cat. No. QT01008651, Qiagen, Hilden, Germany) as an antioxidant marker. The Thunderbird SYBR qPCR Mix Kit (Toyobo, Osaka, Japan) was used for real-time RT-PCR following the manufacturer’s instructions. Furthermore, we performed real-time RT-PCR on ten selected genes to validate the findings of RNA sequencing. To facilitate this process, we designed new primers, as detailed in Appendix A. For normalization purposes, the GAPDH gene was used as a control.

### 4.8. Total RNA Isolation, Library Preparation, and RNA Sequencing for Transcriptome Analysis

The TRIzol reagent (Invitrogen, Waltham, MA, USA) was used to extract total RNA from cells following the manufacturer’s protocol. The extracted total RNA’s quality was assessed using Bioanalyzer 2100 (Agilent, Santa Clara, CA, USA). Total RNA with an RNA Integrity Number (RIN) value of 7 or more was considered for library preparation. Library preparation was performed using the TruSeq Stranded mRNA LT Sample Prep Kit following the manufacturer’s instructions (Illumina, San Diego, CA, USA). The HiSeq X system (Illumina) was used to pair-end sequence nine libraries with their respective indices.

### 4.9. Identification of Differentially Expressed Genes through Mapping of RNA-seq Reads

The sequence data obtained in this study have been deposited in the NCBI SRA database under accession numbers, SRR24775354–SRR24775367. Raw reads were mapped to the GRCh38 reference genome using BBMap with default settings. The number of reads mapped to each transcript was used to identify differentially expressed genes (DEGs) using DESeq2 in DEBrowser v1.24.1 [36]. We compared the Roselle-extract-treated HaCaT cells with those treated with distilled water (control). The number of reads was normalized using the MRE normalization method without any correction. By applying a fold change (FC) more than twice and *p*-values less than 0.05, we identified DEGs for each comparison. For the HS-All comparison, all six datasets from the Roselle-extract-treated samples were compared to the three control datasets.

### 4.10. Functional Characterization of Differentially Expressed Genes (DEGs) by Gene Ontology (GO) Enrichment Analysis

The up-regulated and down-regulated DEGs resulting from HSPE and HSCE treatment were subjected to GO enrichment analysis using the WebGestalt program [37]. This program utilizes an overrepresentation analysis (ORA) against the gene ontology (GO) functional database, which is categorized into three main groups: biological process, cellular component, and molecular function. First, we determined the number of genes mapped to GO Slim. From this, we selected genes annotated to specific functional categories and performed enrichment analysis of the human genome using the following parameters: a minimum number of IDs in a category of 5, and a maximum of 2000 IDs. We used the Bonferroni method for the FDR, and the significance level was set at *p*-value < 0.05 with a top 100 cut-off. Finally, we only considered identified functional gene sets with a *p*-value less than 0.05.

## 5. Conclusions

In this study, we found that water extracts derived from Roselle plants (HSPE) and callus cells (HSCE) have different components, as demonstrated by HPLC analysis. The most common components from both extracts were nucleic acids and metabolites such as phenylalanine and tryptophan. The cell viability assay showed that HSCE is not cytotoxic to HaCaT cells and has strong anti-melanogenic effects. HSCE also has functions for skin barrier and antioxidant activity, but the concentration should be more than 10% to obtain a positive result. Additionally, we conducted transcriptome profiles of HSPE- and HSCE-treated human skin cells for the first time. Our results revealed significant transcriptome differences between human skin cells treated by the two extracts of Roselle. HSPE has more of an effect on human skin cells compared to HSCE. The up-regulated genes by HSPE were induced by hypoxia and function in angiogenesis, the oxidation-reduction process, and glycolysis. In contrast, the up-regulated genes by HSCE encode ribosome proteins of 18S ribosome and IFI6 functioning in the healing of radiation-injured skin cells. Thus, we showed that the two different water extracts from Roselle have different components and effects on human skin cells. Therefore, the commercial uses of the two different water extracts from Roselle for cosmetic and pharmaceutical purposes should be applied differently.

## Figures and Tables

**Figure 1 ijms-24-10853-f001:**
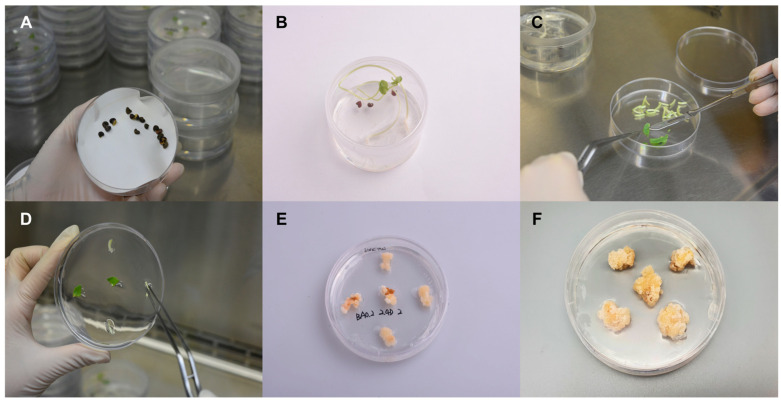
Generation of callus from Roselle seedlings. (**A**) Sterilized Roselle seeds to ensure the absence of contaminants. (**B**) Germination of the Roselle seeds to obtain healthy seedlings. (**C**) The Roselle seedlings cut into small pieces for further tissue culture. (**D**) The Roselle leaves placed on a medium containing Murashige and Skoog (MS) nutrients. (**E**) Induction of callus formation by adding appropriate plant hormones to the medium. (**F**) The development of mature Roselle calluses, which can be used for various applications.

**Figure 2 ijms-24-10853-f002:**
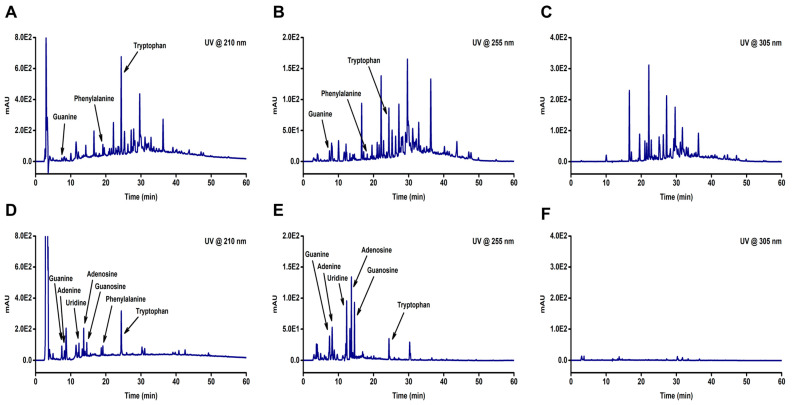
The HPLC chromatograms of two different extracts obtained from *Hibiscus sabdariffa* (HS). The extracts include HSPE, which is the extract obtained from the plant, and HSCE, which is the extract obtained from the callus. The samples were analyzed with a water and acetonitrile mixture containing 0.1% trifluoracetic acid as the mobile phase and detected at 210 nm, 255 nm, and 305 nm using a diode array detector. The chromatograms for (**A**) HSPE (@210 nm), (**B**) HSPE (@255 nm), and (**C**) HSPE (@305 nm) are shown in the top row, while the chromatograms for (**D**) HSCE (@210 nm), (**E**) HSCE (@255 nm), and (**F**) HSCE (@305 nm) are shown in the bottom row.

**Figure 3 ijms-24-10853-f003:**
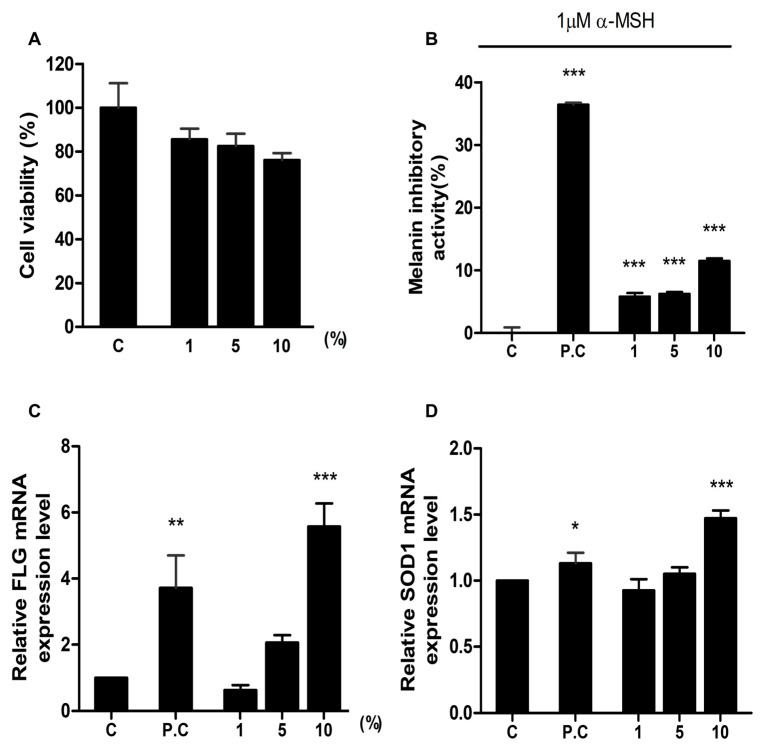
Evaluation of the effects of HSCE on cell cytotoxicity, melanin production, skin barrier, and antioxidant activity. (**A**) Cell viability assay of HaCaT cells treated with HSCE. Inhibition of melanin production by HSCE on B16F1 cells. (**B**) Inhibition of melanin production by HSCE on B16F1 cells. Control without any treatment; positive control (P.C) treated with 100 ppm kojic acid. (**C**) Effects of HSCE on skin barrier function in HaCaT cells. The control group (**C**) was treated with sterile water, and the positive control (P.C) group was treated with 1% glycerol glucoside. (**D**) Antioxidant effect of HSCE on HaCaT cells. The control group (**C**) was treated with sterile water, and the positive control (P.C) group was treated with 2 mM NAC (N-acetylcy-L-cysteine). The gene expression of FLG and SOD1 was analyzed after treatment with three different concentrations of HSCE (1%, 5%, and 10%). The symbols *, **, and *** represent statistical significance at *p* < 0.05, *p* < 0.01, and *p* < 0.001 levels, respectively.

**Figure 4 ijms-24-10853-f004:**
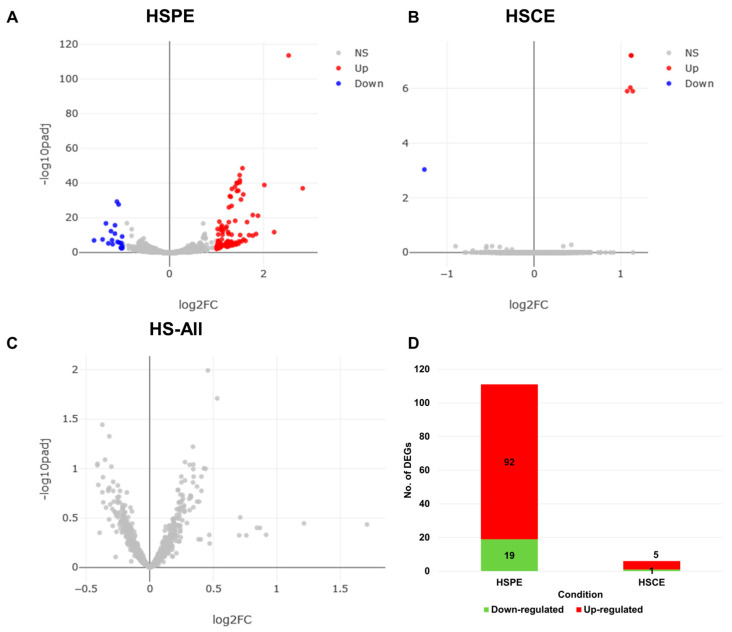
Distribution of differentially expressed RNAs (DERs) and number of DERs in each condition. Volcano plots show the distribution of adjusted *p*-values and fold changes in HSPE (**A**), HSCE (**B**), and HS-All (**C**) conditions. Blue and red dots indicate down- and up-regulated RNAs, respectively. (**D**) The number of identified DERs in HSPE and HSCE conditions.

**Figure 5 ijms-24-10853-f005:**
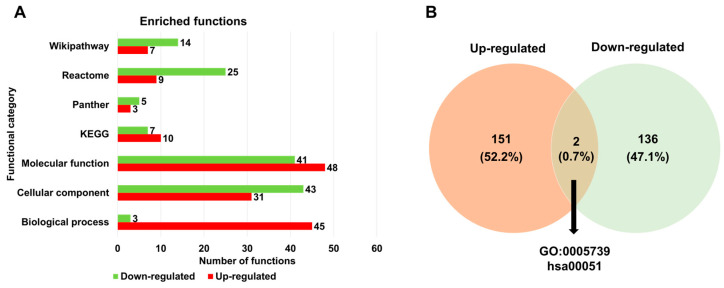
Enriched functions of identified DERs associated with HSPE treatment. (**A**) The number of enriched functions identified for up- and down-regulated DERs based on seven functional categories. (**B**) A Venn diagram displaying the number of enriched functions identified for up- and down-regulated DERs.

**Figure 6 ijms-24-10853-f006:**
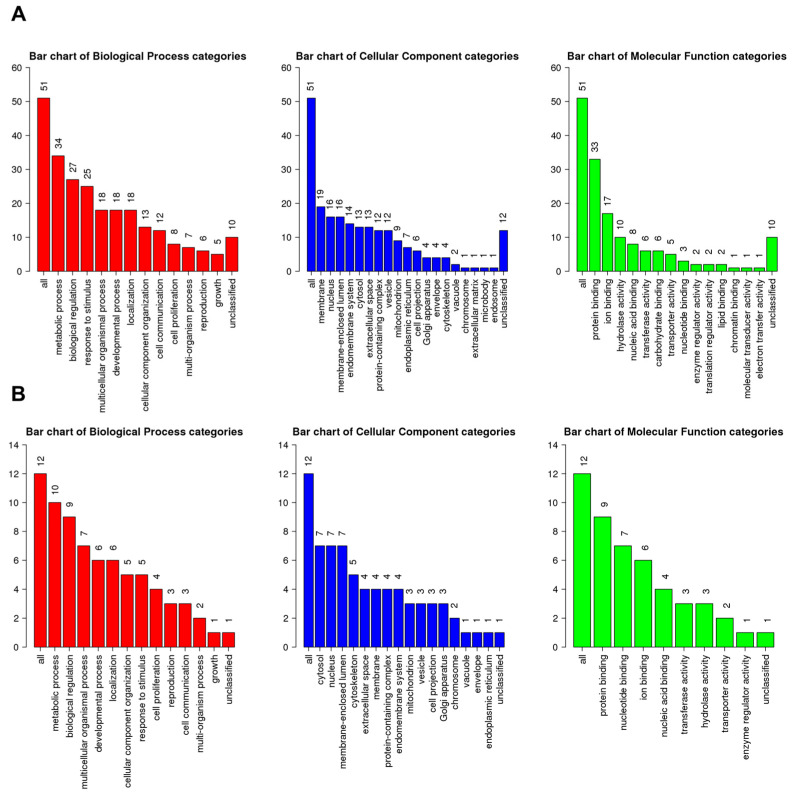
GO slim classification of up- and down-regulated genes by HSPE according to three functional categories: biological process (red), cellular component (blue), and molecular function (green). (**A**) Number of genes assigned to each GO term for up-regulated genes. (**B**) Number of genes assigned to each GO term for down-regulated genes. The numbers indicate the count of genes assigned to the corresponding GO term.

**Figure 7 ijms-24-10853-f007:**
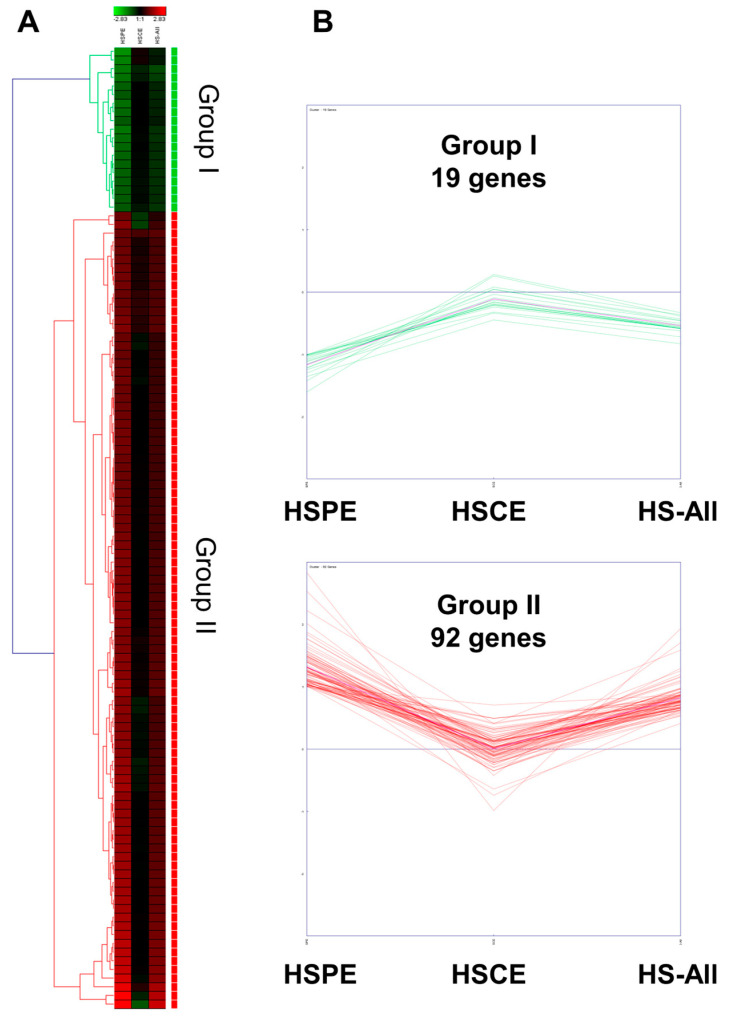
Hierarchical clustering of 111 differentially expressed genes identified from HSPE treatment. (**A**) Heatmap displaying two distinct clusters of 111 differentially expressed genes, defined by hierarchical clustering using the average linkage method, in three different conditions. (**B**) Expression levels of the identified genes in three different conditions: HSPE, HSCE, and HS-All.

**Figure 8 ijms-24-10853-f008:**
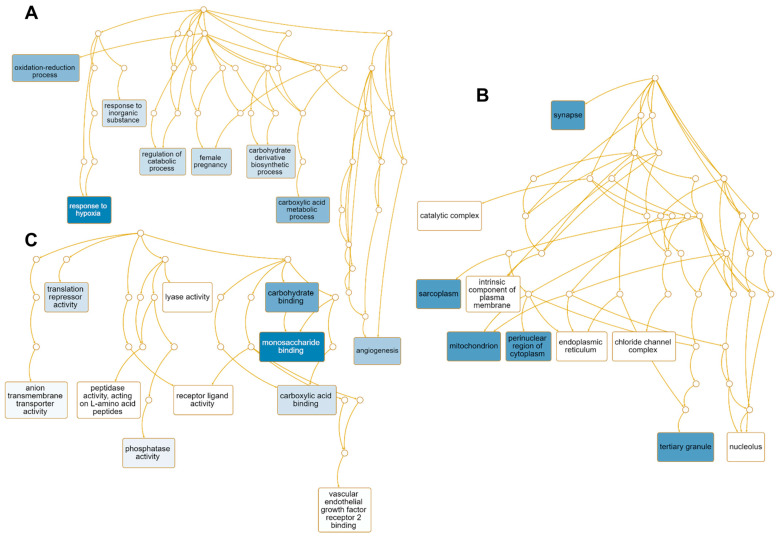
Enriched GO terms associated with 51 up-regulated genes by HSPE treatment. Directed acyclic graphs (DAGs) show the most significant GO terms for up-regulated genes according to biological process (**A**), cellular component (**B**), and molecular function (**C**). The DAG displays the hierarchical relationship of identified GO terms. The blue colored boxes indicate the most significant GO terms. The significance is indicated by the color intensity.

**Figure 9 ijms-24-10853-f009:**
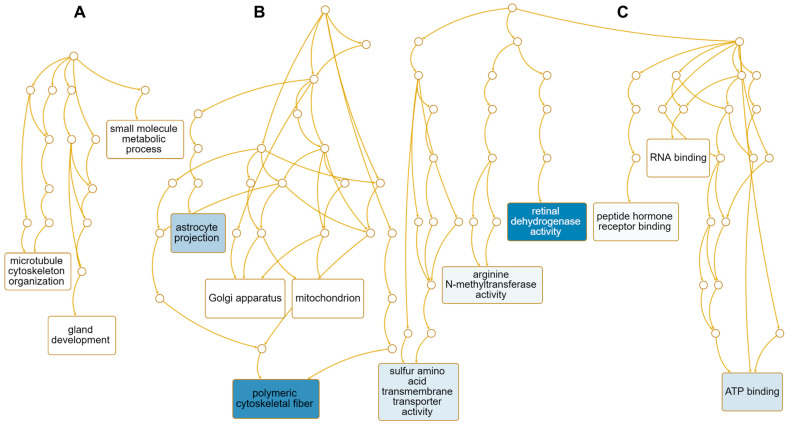
Enriched GO terms associated with 12 down-regulated genes in response to HSPE treatment. Directed acyclic graphs (DAGs) illustrate the most significant GO terms for down-regulated genes in terms of biological process (**A**), cellular component (**B**), and molecular function (**C**). The DAGs show the hierarchical relationship of the identified GO terms, with blue colored boxes indicating the most significant GO terms. The color intensity represents the significance level.

**Figure 10 ijms-24-10853-f010:**
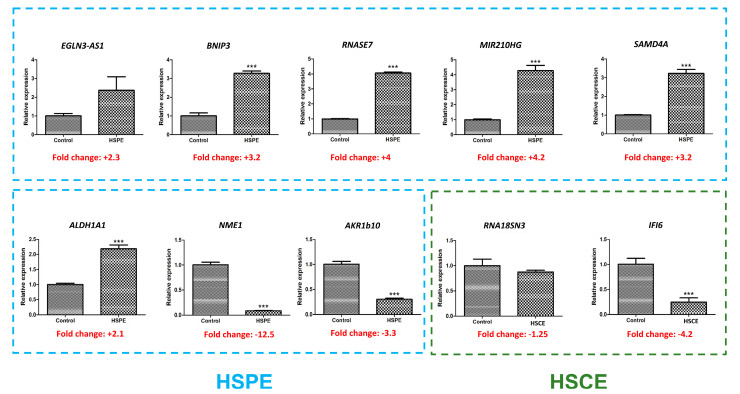
Real-time RT-PCR results showing the expression levels of 10 selected genes. Among these genes, eight (indicated by blue dotted lines) were found to be differentially expressed by HSPE, while two (indicated by green dotted lines) were differentially expressed by HSCE. The presence of ‘***’ indicates a *p*-value less than 0.001, indicating statistical significance in the comparison between the control and treatment groups.

**Table 1 ijms-24-10853-t001:** Details of the nine HaCaT samples used for RNA sequencing.

Index	Condition	Treatment	Library Name	Replicate
1	Control	Distilled water	DW-R1	1
2	Control	Distilled water	DW-R2	2
3	Control	Distilled water	DW-R3	3
4	HSPE	HS extract from plants	HSPE-R1	1
5	HSPE	HS extract from plants	HSPE-R2	2
6	HSPE	HS extract from plants	HSPE-R3	3
7	HSCE	HS extract from callus	HSCE-R1	1
8	HSCE	HS extract from callus	HSCE-R2	2
9	HSCE	HS extract from callus	HSCE-R3	3

**Table 2 ijms-24-10853-t002:** Summary of raw RNA sequencing data obtained.

Library Name	Total Read (bp)	Total Reads	GC(%)	AT(%)	Q20(%)	Q30(%)
DW-R1	10,517,649,342	104,135,142	40.2	59.8	98.5	95.5
DW-R2	13,311,500,636	131,797,036	41.1	58.9	98.6	95.7
DW-R3	11,736,440,986	116,202,386	43	57	98.5	95.6
HSPE-R1	10,081,467,712	99,816,512	41.2	58.8	98.5	95.7
HSPE-R2	10,758,263,258	106,517,458	40.8	59.2	98.5	95.4
HSPE-R3	13,290,280,334	131,586,934	42.2	57.8	98.5	95.6
HSCE-R1	13,310,397,918	131,786,118	43.5	56.5	98.5	95.7
HSCE-R2	10,762,588,482	106,560,282	42.3	57.7	98.7	96.1
HSCE-R3	13,324,897,276	131,929,676	40.8	59.2	98.7	95.9

Abbreviations used: bp, base pairs; Q20 and Q30, quality scores of 20 and 30, respectively, for raw sequencing data.

**Table 3 ijms-24-10853-t003:** Ten representative genes differentially expressed by HSPE treatment.

Gene Name	NCBI ID	Description	Padj	log_2_FC
EGLN3-AS1	XR_943736.2	EGLN3 antisense RNA 1, ncRNA	9.76 × 10^−38^	2.832333775
BNIP3	NM_004052.3	BCL2 interacting protein 3	2.64 × 10^−114^	2.533223896
RNASE7	NM_032572.3	ribonuclease A family member 7	1.85 × 10^−12^	2.225154471
MIR210HG	NR_038262.1	MIR210 host gene, long non-coding RNA	1.27 × 10^−39^	2.016904685
SAMD4A	NM_015589.5	sterile alpha motif domain containing 4A	1.77 × 10^−10^	1.770832281
PPP1R3C	NM_005398.6	phosphatase 1 regulatory subunit 3C	2.11 × 10^−11^	1.840645949
SLC6A8	NM_001142806.1	solute carrier family 6 member 8	2.60 × 10^−22^	1.771043453
ALDH1A1	NM_000689.4	aldehyde dehydrogenase 1 family member A1	5.40 × 10^−6^	−1.302747218
NME1	NM_000269.2	NME/NM23 nucleoside diphosphate kinase 1	1.56 × 10^−17^	−1.355005295
AKR1B10	XM_011516416.1	aldo-keto reductase family 1 member B10	1.04 × 10^−7^	−1.607186598

Abbreviations: padj, adjusted *p*-value; log_2_FC, log_2_ converted fold change.

**Table 4 ijms-24-10853-t004:** RNAs differentially expressed following HSCE treatment.

Gene Name	NCBI ID	Description	Padj	log_2_FC
IFI6	NM_022873.2	interferon alpha inducible protein 6	0.000925	−1.26271
RNA18SN2	NR_146146.1	18S ribosomal N2	1.27 × 10^−6^	1.072622
RNA18SN4	NR_146119.1	18S pre-ribosomal N4	9.39 × 10^−7^	1.109612
RNA18SN1	NR_145820.1	18S ribosomal N1	6.30 × 10^−8^	1.116593
RNA18SN5	NR_003286.4	18S ribosomal N5	6.30 × 10^−8^	1.121973
RNA18SN3	NR_146152.1	18S ribosomal N3	1.27 × 10^−6^	1.136712

Abbreviations: Adjusted *p*-value (padj), Log_2_-fold change (log_2_FC).

**Table 5 ijms-24-10853-t005:** Selected genes for real-time RT-PCR analysis and RNA sequencing results.

Condition	Gene Name	NCBI Reference ID	Description	Padj	log_2_FC
HSPE	*EGLN3-AS1*	XR_943736.2	EGLN3 antisense RNA 1, ncRNA	9.76 × 10^−38^	2.832
HSPE	*BNIP3*	NM_004052.3	BCL2 interacting protein 3	2.64 × 10^−114^	2.533
HSPE	*RNASE7*	NM_032572.3	ribonuclease A family member 7	1.85 × 10^−12^	2.225
HSPE	*MIR210HG*	NR_038262.1	MIR210 host gene, long non-coding RNA	1.27 × 10^−39^	2.017
HSPE	*SAMD4A*	NM_015589.5	sterile alpha motif domain containing 4A	1.77 × 10^−10^	1.771
HSPE	*ALDH1A1*	NM_000689.4	aldehyde dehydrogenase 1 family member A1	5.40 × 10^−6^	−1.3
HSPE	*NME1*	NM_000269.2	NME/NM23 nucleoside diphosphate kinase 1	1.56 × 10^−17^	−1.36
HSPE	*AKR1B10*	XM_011516416.1	aldo-keto reductase family 1 member B10	1.04 × 10^−7^	−1.61
HSCE	*IFI6*	NM_022873.2	interferon alpha inducible protein 6	0.0009255	−1.26
HSCE	*RNA18SN3*	NR_146152.1	18S ribosomal N3	1.27 × 10^−6^	1.137

Abbreviations: Adjusted *p*-value (padj), Log_2_-fold change (log_2_FC).

## Data Availability

The raw data were deposited in the NCBI SRA database with the following accession numbers: SRR24775354–SRR24775367.

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
