# Peer review of "Comparative Analysis of Water Extracts from Roselle (Hibiscus sabdariffa L.) Plants and Callus Cells: Constituents, Effects on Human Skin Cells, and Transcriptome Profiles"

_ijms, 2023, doi:10.3390/ijms241310853_

Round 1

Author Response

Reviewer #1

In this article Won Kyong Cho et al., investigated the potential of water extracts derived from roselle leaves and callus cells for cosmetics and pharmaceutical purposes. They generated callus from roselle leaves and produced two different water extracts through heat extraction, and named it Hibiscus sabdariffa plant extract (HSPE) and Hibiscus sabdariffa callus extract (HSCE) and then further analyzed it using different techniques for cosmetic and pharmaceutical application. The study is good but there are some points which needs to be clarified which are given below.

Comments to Authors

Comment 1. In the introduction section the authors mostly focused on the importance of the flowers extract of Roselle but did not mention that why they choose leaves for this study not the flowers?

Response: We have addressed why we selected leaves instead of Roselle flowers as follows.

In the manuscript

L70-78

In this study, we selected on leaves of Roselle because generating callus from Roselle flowers can be more challenging compared to using leaves. Callus formation depends on various factors, including the tissue's ability to undergo dedifferentiation and reprogram into a totipotent state. In many plant species, including Roselle, leaves generally have a higher dedifferentiation capacity compared to other plant organs. Flower tissues, such as petals or reproductive structures, have a specialized cellular composition and are more differentiated than leaves. These specialized cells may be less responsive to in vitro culture conditions and callus induction due to developmental changes. Therefore, we decided to study leaves as they have a higher propensity for callus formation in this study.

Comment 2. It is not mentioned anywhere that why heat extraction method was chosen over other methods?

Response: We have addressed why we selected heat extraction method in the manuscript.

In the manuscript

L99-103

The heat extraction method was chosen based on the intended application of the Roselle extract, which is being developed as a sterilized additive for inclusion in cosmetics and non-prescription pharmaceuticals. The selection of this method aligns with the manufacturing process of these products and consequently led to the utilization of high-pressure sterilization extraction.

Comment 3. The passage time of the cells should also be mentioned in the experimental section as it may impact the subsequent results.

Response: We have addressed the passage time of HaCaT cells as follows:

In the manuscript

L555-556

We conducted the experiments using HaCaT cells that had undergone passages ranging from 10 to 15 or fewer.

Comment 4. Line number “566” the word “by” is written two times. Please delete one of them.

Response: We have revised it.

Comment 5. Why UHPLC-QTOF and LC-MS were not used for qualitative and quantitative analysis instead of HPLC, respectively. As this is an important point which may affect the later results.

Response: In this study, the use of HPLC was not intended for individual peak identification and quantification of substances present in the plant and callus extracts. Instead, it aimed to determine whether there were any differences in the chromatographic patterns between the two samples. Particularly, since the samples were "young leaves" and "callus derived from young leaves," it was predicted that their chromatographic patterns would be somewhat similar. However, the analysis revealed completely different patterns. Additionally, although several amino acids and nucleic acid-related substances were observed in the chromatograms, their mention was omitted in the main text. Nevertheless, they were identified by matching the data obtained from the internally established ESI-MS/MS with the components.

Comment 6. All the activities are determined mostly using a single cell type such as in case of cell viability or skin barrier or antioxidant effects? Is it enough to reach the conclusion based on single type of in vitro experiments? And also 10% concentration isn’t too high for obtaining these effects?

Response: While it is common to perform in vitro evaluations of skin barrier and antioxidant effects using a specific cell type, such as the HaCaT cell line, there are considerations to keep in mind regarding drawing conclusions solely from single-type in vitro experiments. Although these experiments provide valuable insights, it is important to validate the findings using multiple approaches and cell types to ensure the robustness and generalizability of the conclusions.

Regarding the 10% concentration, which may seem high, it is crucial to consider the final effective concentration applied to the cells. In this case, the 10% concentration corresponds to a final concentration of 0.02% when prepared at a ratio of 2g/L. Evaluating the effects at this concentration allows for a suitable assessment of the extract's potential effects while considering practical application and potential dilution factors. We have addressed about 10% concentration in the manuscript.

In the manuscript

L559-562

In this case, the 10% concentration corresponds to a final concentration of 0.02% when prepared at a ratio of 2g/L. Evaluating the effects at this concentration allows for a suitable assessment of the extract's potential effects while considering practical application and potential dilution factors.

Comment 7. As the authors discussed about the importance of solvent for extraction and have pointed out that most of the hydrophobic compounds cannot be extracted completely using water then why does not authors selected any organic solvent for comparative purposes to see what changes and effects it will have on the extracts composition and function? I think it would have made the study more interesting.

Response: The intention of this study is to develop cosmetics materials using Roselle as the raw material. The content of this paper focuses on verifying the efficacy of HSPE or HSCE at various stages of the overall research. In general, when adding ingredients to cosmetics, consumers prefer products that do not contain organic solvents like alcohol. Therefore, even cosmetic manufacturers prefer to use materials without organic solvents or, if used, only in very limited quantities. As a result, we conducted our research by extracting using water.

Reviewer 2 Report

The authors report on a comparative investigation of two water extracts of roselle. I only have a few smaller comments/requests:

-How were the roselle seeds sterilized? Would the sterilization method interfere with viability/activity of the extracts?

-Fig. 3: Obviously, these results were done on HSCE. How about HSPE? Did you also investigate on HSPE and if yes, can you say something about the results?

-Fig. 3: The increase of the melanin inhibition effect in 3B is about in the same range as the cytotox effect in 3A. Could the effect in 3B just be a cytotox effect?

-Fig. 3D: the fold change of SOD1 does not seem very high (below 1.5fold). Is it relevant? In the transcriptomics analysis further down in the manuscript, you apply a change of 2fold as relevant. Can you comment?

-Results: The transcriptomics do not show any change in melanogenesis pathway nor e.g. barrier maturation. This is somewhat surprising given the results in Fig. 3 with melanin inhibition and FLG expression. Can you comment on this?

-Discussion:

-line 408-410: These two sentences read like a contradiction. Can you please clarify.

-line 418-421: Again here, first you say it is increased, then you say it needs more than 10% for a positive results. It again seems like a contradiction. Can you please clarify.

-In general, the Discussion part is too long and contains a lot of speculation that is not really relevant to the results found. I think you could shorten this for better reading. However, the relevant finding that the results of Fig. 3 are not found in the transcriptomics analysis is missing. This should be included and discussed in the Discussion section.

Materials and Methods:

-there is a typo in line 566: one 'by' too much.

Author Response

Reviewer #3

The authors report on a comparative investigation of two water extracts of roselle. I only have a few smaller comments/requests:

Comment 1: How were the roselle seeds sterilized? Would the sterilization method interfere with viability/activity of the extracts?

Response: Thank you for your question regarding the sterilization of roselle seeds. The seeds are sterilized using a solution of 70% ethanol and 2.5% sodium hypochlorite (NaOCl). The sterilization process involves subjecting the seeds to ethanol for 1 minute and sodium hypochlorite for 20 minutes. This method is carefully designed to minimize any potential interference with the viability and activity of the extracts. Additionally, Tween-20 can be added to enhance the sterilization effect. After sterilization, the seeds are thoroughly rinsed with sterilized water multiple times to ensure complete removal of the sterilization solution.

We have revised the associated sentences as follows.

In the manuscript

L509-513

To sterilize the seeds, 25 seeds were treated with 70% EtOH for 1 minute, followed by 2.5% NaOCl for 20 minutes. This method is carefully designed to minimize any potential inter-ference with the viability and activity of the extracts. Additionally, Tween-20 can be added to enhance the sterilization effect. After sterilization, the seeds are thoroughly rinsed with sterilized water multiple times to ensure complete removal of the sterilization solution.

Comment 2: Fig. 3: Obviously, these results were done on HSCE. How about HSPE? Did you also investigate on HSPE and if yes, can you say something about the results?

Response: Thank you for your comment and question. Regarding HSPE, we did not specifically conduct in vitro efficacy evaluations for this extract. Our main focus was on HSCE, where we conducted comprehensive in vitro efficacy evaluations and performed transcriptome analysis to support the obtained data. The analysis included both callus and plant samples, enabling a comparison between the two. Thus far, we do not have any results from the in vitro efficacy evaluations of HSPE.

Comment 3: Fig. 3: The increase of the melanin inhibition effect in 3B is about in the same range as the cytotox effect in 3A. Could the effect in 3B just be a cytotox effect?

Response: The cell viability results in Fig. 3A are obtained from HaCaT cells, which are keratinocytes. The melanin inhibition results in Fig. 3B are obtained from B16F1 cells, which are melanocytes. After reviewing the completed efficacy evaluation data using HSCE samples, we did not conduct a separate test for cytotoxicity in melanocytes. Since the target cell lines are different, it is difficult to speculate that the slight decrease in keratinocyte cell viability in Fig. 3A is directly related to melanin inhibition. Therefore, it is not likely that the effect observed in Fig. 3B is solely a cytotoxic effect.

In addition, we have indicated cell line names in the legend for Figure 3 as follows.

In the manuscript

L132-138

(A) Cell viability assay of HaCaT cells treated with HSCE. Inhibition of melanin production by HSCE on B16F1 cells. (B) Inhibition of melanin production by HSCE on B16F1 cells. Control without any treatment, positive control (P.C) treated with 100 ppm kojic acid. (C) Effects of HSCE on skin barrier function in HaCaT cells. The control group (C) was treated with sterile water, and the positive control (P.C) group was treated with 1% glycerol glucoside. (D) Antioxidant effect of HSCE on HaCaT cells. The control group (C) was treated with sterile water, and the positive control (P.C) group was treated with 2 mM NAC (N-acetylcy-L-cysteine).

Comment 4: Fig. 3D: the fold change of SOD1 does not seem very high (below 1.5fold). Is it relevant? In the transcriptomics analysis further down in the manuscript, you apply a change of 2fold as relevant. Can you comment?

Response: Real-time RT-PCR and RNA-sequencing are both used to analyze gene expression levels, but these two methods have several differences, including the normalization process. Therefore, it is more common to compare and analyze the patterns of gene expression, such as up-regulation or down-regulation, rather than expecting the same fold change values from both methods. Additionally, the samples used in Figure 3 and the samples used in RNA-sequencing are independent biological samples, which may exhibit slight differences. These aspects can be found in many other references. As mentioned in our materials and methods, for RNA sequencing, the significance of the data is indicated by a minimum of P < 0.05 and a maximum of P < 0.001, while for the mentioned data, the significance is P < 0.01. Therefore, we understand your concern, but as mentioned earlier, the trend of fold values is similar, and the significance is determined based on the p-values, which confirms the validity of the analysis.

Comment 5: Results: The transcriptomics do not show any change in melanogenesis pathway nor e.g. barrier maturation. This is somewhat surprising given the results in Fig. 3 with melanin inhibition and FLG RNA expression by real time RT-PCR. Can you comment on this?

Response: The lack of observed changes in the melanogenesis pathway and barrier maturation in the transcriptomics data, despite the results in Figure 3 showing melanin inhibition and FLG RNA expression by real-time RT-PCR, may appear surprising. It is important to consider that transcriptomics data provides information on gene expression levels, while functional outcomes such as melanin inhibition and FLG expression are influenced by various factors beyond gene expression alone.

The regulation of melanogenesis and barrier maturation is a complex process that involves multiple factors, including post-transcriptional and post-translational modifications, protein interactions, and cellular signaling pathways. These aspects may not be fully captured in the transcriptomics data, which primarily focuses on gene expression.

It is also important to note that real-time RT-PCR provides a more targeted and specific analysis of gene expression compared to transcriptomics, which surveys the entire transcriptome. The results obtained from real-time RT-PCR in Figure 3 reflect the expression of specific genes related to melanin inhibition and FLG, but they may not represent the overall changes in the transcriptome.

-Discussion:

Comment 6: line 408-410: These two sentences read like a contradiction. Can you please clarify.

Response: As per the reviewer's comment, the paragraph might be confusing for readers. Consequently, we have made revisions to the paragraph as follows:

In the manuscript:

L403-410

Normally, only 1% of extracts derived from diverse plant cells are commercially utilized. However, in this study, we investigated the effects of three different concentrations, specifically 1%, 5%, and 10%. The results from the cell viability assay conducted on HaCaT cells indicate that the water extract of roselle cells exhibited no cytotoxicity, irrespective of the HSCE concentration used. Furthermore, the water extract showed lower cytotoxicity com-pared to the control group treated with sterile water. Interestingly, the cell viability remained unaffected by the concentration of HSCE in the water extract of roselle cells.

Comment 7: line 418-421: Again here, first you say it is increased, then you say it needs more than 10% for a positive results. It again seems like a contradiction. Can you please clarify.

Response: As per the reviewer's comment, the paragraph might be confusing for readers. Consequently, we have made revisions to the paragraph as follows:

In the manuscript:

L417-424

It was observed that only the highest concentration of HSCE (10%) resulted in a significant difference (1.47) when compared to the control group. This implies that the 10% HSCE treatment exhibited a more pronounced antioxidant effect in terms of SOD1 gene expression. However, it is worth noting that there was an overall increase in SOD1 expression after treatment with both 5% and 10% HSCE, albeit without reaching statistical significance. These results suggest that HSCE, particularly at a concentration of 10%, has the potential to induce antioxidant effects through the modulation of SOD1 gene expression in HaCaT cells.

Comment 8: In general, the Discussion part is too long and contains a lot of speculation that is not really relevant to the results found. I think you could shorten this for better reading. However, the relevant finding that the results of Fig. 3 are not found in the transcriptomics analysis is missing. This should be included and discussed in the Discussion section.

Response: Following the reviewer's suggestion, we have removed irrelevant paragraphs from the study. We retained the discussion sections that focus on the novel findings of differentially expressed genes identified through RNA-sequencing. Additionally, we have addressed the disparity between the results obtained from real-time RT-PCR and RNA-sequencing in the following manner.

In the manuscript

L484-495

Significant differences were observed in the findings obtained from real-time RT-PCR analysis and transcriptome analysis using RNA-sequencing when examining the effects of Roselle plant cell extracts. Specifically, real-time RT-PCR analysis yielded specific re-sults related to cytotoxicity, FLG expression, SOD1 expression, and α-MSH expression following HSCE treatment. In contrast, transcriptome analysis provided comprehensive insights into the impact of HSPE and HSCE on HaCaT cells, highlighting variations in gene expression patterns and their connections to diverse biological processes. Real-time RT-PCR and RNA-sequencing are complementary techniques, each with its own strengths and limitations. Due to disparities in sensitivity, targeted versus unbiased approach, technical variability, data analysis, and biological variability, the specific outcomes de-rived from real-time RT-PCR may not directly align with those obtained through tran-scriptome analysis via RNA-sequencing.

Materials and Methods:

Comment 9: there is a typo in line 566: one 'by' too much.

Response: We have revised it.

Round 2

Reviewer 1 Report

The paper has been improved.

Reviewer 2 Report

Thank you for the revised version of the manuscript. It is fine for me in the present form.